

# Adaptive gamification in collaborative virtual classroom: a systematic review

Intan Yusrina Zairon, Tengku Siti Meriam Tengku Wook, Syahanim M. Salleh and Hadi Affendy Dahlan

Faculty of Information Science and Technology, Universiti Kebangsaan Malaysia, Bangi, Selangor, Malaysia

## ABSTRACT

This study examines the potential of adaptive gamification in tackling challenges in collaborative virtual classrooms, including sustaining engagement, fostering motivation, and enhancing teamwork. The research identifies key theories and frameworks essential for designing gamified virtual learning environments by employing a systematic review guided by the Preferred Reporting Items for Systematic Reviews and Meta-Analyses (PRISMA) framework. The methodology involved a four-phase process: identification, screening, eligibility assessment, and inclusion of relevant studies published between 2019 and 2024 in databases such as Scopus, Web of Science, IEEE, and ScienceDirect. A total of 43 articles were analyzed to derive themes and insights. The findings emphasize the integration of motivational frameworks like Self-Determination Theory (SDT) and learning models such as constructivism to enhance learner engagement and academic performance. These theories, centered on autonomy, competence, and relatedness, are effectively supported through adaptive gamification strategies. Frameworks like the Felder-Silverman learning style model (FSLSM) enable personalization by aligning gamified content with individual learning preferences, improving motivation and inclusivity. Furthermore, collaborative learning theories, such as online collaborative learning (OCL), provide a foundation for designing environments that promote peer interaction, mutual accountability, and teamwork. These frameworks balance individual and collective goals, transforming online education into an engaging and collaborative experience. This review concludes that adaptive gamification, underpinned by strong theoretical and systematic analysis, has significant potential to enhance virtual education by creating dynamic, personalized, and inclusive learning spaces that address diverse learner needs.

## INTRODUCTION

In recent years, virtual learning environments (VLEs) have become an integral part of contemporary education, driven by the growing emphasis on digital and remote learning. The platforms have revolutionized the delivery of education, offering flexibility and accessibility to learners worldwide. However, alongside their advantages, VLEs present significant challenges in sustaining engagement, motivation, and collaborative

Corresponding author
Intan Yusrina Zairon,
intanyusrina@gmail.com

participation which are the essential factors for meaningful learning experiences. Therefore, addressing these challenges has become a critical focus for educators and researchers alike. One innovative approach gaining traction is gamification, which involves integrating game-like elements such as points, badges, leaderboards, and achievement rewards into educational settings to enhance participation and improve the overall learning experience. Research indicates that gamification can significantly boost student motivation, engagement and interactivity by introducing goal-oriented tasks and fostering a sense of accomplishment (*Hellín et al., 2023*; *Jayalath & Esichaikul, 2022*; *Wook et al., 2021*). Despite these advantages, traditional gamification models often prioritize competitive dynamics, which, while effective for individual motivation, can inadvertently discourage peer collaboration which is a cornerstone of effective learning, particularly in virtual classrooms.

Adaptive gamification offers a promising approach to tackle these challenges by personalizing learning experiences to align with individual learner preferences while incorporating game mechanics that promote active participation and collaborative skills (*Lavoué et al., 2018*; *Zourmpakis, Kalogiannakis & Papadakis, 2023*). This personalized approach creates an autonomous learning environment where engagement is sustained through continuously optimized challenges and rewards, effectively balancing both intrinsic and extrinsic motivational factors. By dynamically tailoring gamified elements to individual learner preferences, adaptive gamification strikes a balance between competition and collaboration. This approach fosters a supportive and engaging environment where learners are motivated not only to achieve personal goals but also to contribute to group success. In the context of collaborative virtual classrooms, adaptive gamification holds the potential to enhance both individual and collective learning experiences by encouraging teamwork, mutual accountability, and shared problem-solving (*Hallifax et al., 2019a*; *Mohamad et al., 2019*).

The theoretical underpinning of adaptive gamification draws from various educational theories and psychological frameworks. For instance, the Self-Determination Theory (SDT) provides insights into how autonomy, competence, and relatedness drive intrinsic motivation (*Ryan & Deci, 2020*), while the Flow Theory explains how optimal challenge levels maintain engagement (*Abd El-Sattar, 2023*). Additionally, the Felder-Silverman learning style model offers a structured approach to understanding and accommodating diverse learning preferences within a gamified environment (*Bennis, Kandali & Bennis, 2022*; *Syazwani, Noor & Mohamed, 2018*; *Zaric & Scepanovic, 2018*). These theoretical foundations are crucial in developing adaptive systems that can modify instructional content and motivational elements based on real-time learner data. Meanwhile, the significance of peer interaction and knowledge co-construction becomes essential in virtual collaborative settings. Social constructivism and collaborative learning theories suggest that knowledge is best constructed through social interaction and peer collaboration (*Herrera-Pavo, 2021*; *Weinberger & Shonfeld, 2020*; *Zhang, Wen & Liu, 2022*). Adaptive gamification mechanisms can be strategically deployed to encourage meaningful collaboration through carefully designed reward structures that recognize both individual contributions and collective achievements. The implementation of dynamic

collaborative challenges and team-based quests that leverage diverse learning styles and competencies fosters a learning environment where individual accountability and collective responsibility are harmoniously balanced.

Despite the theoretical promise of adaptive gamification, its practical application within collaborative virtual classrooms remains largely unexplored in contemporary educational research. While extensive literature exists on static gamification in individual learning contexts, there is a notable gap in understanding how different theoretical frameworks can be integrated to develop effective adaptive gamification systems in collaborative virtual learning environments. This systematic review aims to bridge this knowledge gap by conducting a comprehensive analysis of relevant theories and their potential applications in adaptive gamification strategies. This research is particularly significant as it seeks to establish a theoretical foundation for developing adaptive gamification frameworks in virtual learning environments. By examining various learning theories, motivational models, and gamification principles, this study aims to identify the most suitable theoretical frameworks that can guide the development of effective adaptive gamification systems. Understanding the theoretical foundations that can effectively support adaptive gamification implementation is crucial for developing robust and effective virtual education systems. The findings will contribute to the growing body of knowledge in educational technology and provide theoretical insights for researchers and practitioners working on adaptive gamification implementations in virtual learning environments.

This systematic review will not only synthesize existing theoretical frameworks but also propose how these theories can be integrated to create comprehensive adaptive gamification models that enhance learning experiences while maintaining individual engagement and motivation in virtual learning environments. To achieve these objectives, this research addresses the following research questions:

- RQ1: How does gamification influence learner motivation and engagement in virtual learning environments?
- RQ2: How can gamification be tailored to individual learners to enhance engagement and motivation in virtual collaborative classrooms?
- RQ3: How can collaborative learning and engagement among learners in virtual classrooms be enhanced with adaptive gamification?

## Background

Adaptive gamification represents a dynamic and personalized approach to designing gamified systems that cater to diverse user preferences, needs, and behaviors (_Tenório et al., 2022_). Traditional gamification methods typically employ static game mechanics such as points, badges, and leaderboards uniformly across all users, often neglecting individual differences in motivation, engagement, and learning styles. This "one-size-fits-all" approach can limit the effectiveness of gamification, particularly in contexts where users exhibit varying skill levels, interests, and goals (_Rodrigues et al., 2021_). The adaptive gamification model addresses these limitations by utilizing mechanisms that dynamically

adjust game elements based on real-time data and user feedback, creating a more responsive and inclusive experience.

A fundamental aspect of adaptive gamification is user modeling, which involves collecting and analyzing user data to create detailed profiles. The design and implementation of adaptive gamification are grounded in several key theories and models, which can be categorized into three thematic areas: motivation, learning style and collaboration. SDT (*Deci et al., 1985*) is central to understanding user motivation in adaptive gamification. SDT emphasizes the importance of autonomy, competence, and relatedness in fostering intrinsic motivation. By aligning gamified elements with these psychological needs, adaptive systems can enhance user engagement and sustain motivation over time. Meanwhile, Online Collaborative Learning (OCL) theory (*Harasim, 2015*) highlights the importance of social interaction and group dynamic in virtual learning environment also results in enhancing both engagement and knowledge retention. Besides, the Felder-Silverman learning styles model (FSLSM) (*Felder & Silverman, 1988*) provides a framework for understanding individual differences in learning preferences. By leveraging this information, the model aligns game mechanics with the unique characteristics of each user. Research by *Hallifax et al. (2019b)* and *Dalponte Ayastuy, Torres & Fernández (2021)* indicates that adaptive gamification systems utilize student profiles to tailor aspects such as difficulty levels, feedback, and rewards, thereby enhancing engagement and motivation. This personalization ensures that learners interact with the system at a level suited to their abilities, leading to optimized learning experiences. For instance, competitive users may benefit from leaderboard-based challenges, while those who prefer collaboration may engage more effectively through cooperative tasks and team-based rewards. This personalization keeps gamified experiences relevant and engaging, fostering sustained motivation over time.

Another key component of adaptive gamification is its contextual adaptability. Beyond understanding user preferences, the model considers the specific context in which gamification is applied. This includes task characteristics, goal complexity, and overall system objectives. In educational settings, adaptive gamification may modify challenge difficulty or feedback types based on student progress and performance. Similarly, in healthcare applications, the model can personalize motivational prompts and rewards to encourage adherence to treatment plans or healthy behaviors.

Additionally, the design framework of adaptive gamification plays a crucial role in the effectiveness of these systems. *Böckle et al. (2018)* propose a design framework that integrates personalized incentive mechanisms, emphasizing the importance of aligning game elements with users' unique traits. This aligns with findings by *Kaeophanuek & Chaisriya (2022)* who advocate for the use of intelligent learning systems, such as machine learning and artificial intelligence (AI), to track student progress and dynamically adjust gamified content. Studies by *Lopez & Tucker (2021)* highlight the potential of machine learning models to predict student performance and tailor gamification strategies to enhance both engagement and knowledge retention. This adaptability is particularly valuable in diverse educational environments, where learners may differ significantly in pace, motivation, and learning styles. In medical education, key design principles have

been identified to tailor gamification strategies that address the unique challenges faced by medical students, including real-time feedback and context-based challenges (*Wang, Kong & Wang, 2024*).

One of the major advantages of adaptive gamification is its potential to improve outcomes across various domains. In education, personalized gamification strategies can enhance student engagement, comprehension, and retention by aligning activities with individual learning styles (*Borotić & Jagušt, 2022*; *Hassan et al., 2021*; *Ibisu, 2024*; *Rodrigues et al., 2021*). Meanwhile, in healthcare, adaptive gamification can motivate patients to adhere to treatment regimens and adopt healthier lifestyles by providing customized rewards and feedback (*Carlier, De Backere & De Turck, 2024*; *Martinho et al., 2020*). In corporate training, this model can meet the diverse needs of employees by tailoring learning modules and gamification incentives, thereby improving skill development and productivity (*Larson, 2020*).

Despite its significant potential, the development and implementation of adaptive gamification models pose several challenges. As highlighted by *Maher, Moussa & Khalifa (2020)*, the continuous need to collect and analyze user data raises ethical concerns regarding privacy and data security. Designing an effective adaptation engine requires a deep understanding of user behavior, robust data collection mechanisms, and advanced computational techniques. Another challenge lies in balancing personalization with consistency, as overly individualized experiences may compromise the overall coherence of the system design (*John et al., 2024*; *Zhao, 2024*). Additionally, reliance on advanced technologies such as machine learning and AI may not be feasible in all educational settings, particularly those with limited resources. Ensuring accessibility while maintaining personalization is also a critical issue, as overly complex gamification systems may become difficult for certain learner groups to navigate. Moreover, the novelty of adaptive gamification may diminish over time, potentially leading to decreased motivation (*Heilbrunn, Herzig & Schill, 2014*). This underscores the need for continuous innovation in maintaining engaging gamified experiences that remain fresh and stimulating.

As virtual learning environments continue to evolve, adaptive gamification presents a promising approach to enhance learner motivation, engagements, and collaboration. By integrating theoretical frameworks such as SDT, FSLSM, and OCL, this review highlights the potential of adaptive gamification to support personalized and collaborative learning experiences. The findings underscore the importance of aligning motivational frameworks, learning models, and adaptive strategies with user needs, while addressing challenges to ensure sustained engagement and motivation in virtual learning environments.

## MATERIALS AND METHODS

This section outlines the systematic review methodology employed to investigate adaptive gamification in collaborative virtual classrooms. To ensure a comprehensive and transparent approach, the review was conducted following the Preferred Reporting Items for Systematic Reviews and Meta-Analyses (PRISMA) guidelines, which provide a structured framework for systematically identifying, evaluating, and synthesizing relevant research. The methodology involved four key phases: (1) identification of relevant studies

**Table 1 Keywords for database search.**

| Database | Keywords |
|---|---|
| Scopus | TITLE-ABS-KEY ("adapt*" OR "personalize*") AND ("virtual learn*" OR "virtual classroom" OR "vr") AND gam* AND "learn* theory" OR "constructiv*" OR "online collaborat* learning" OR "motivat* theory") AND learning AND style AND (LIMIT-TO (DOCTYPE, "ar")) AND (LIMIT-TO (SRCTYPE, "j")) **Date of Access: May 2024** |
| WoS | ALL = (("adapt*" OR "personal*") AND ("virtual learn*" OR "virtual classroom" OR "vr") AND gami* AND ("learn* theory" OR "online collaborative learning" OR "engage*" OR "motivat* theory") and 2018 or 2019 or 2020 or 2021 or 2022 or 2023 or 2024 (Publication Years) and 2024 (Exclude—Publication Years) and Article (Document Types) and English (Languages) **Date of Access: May 2024** |
| IEEE | (("adapt*" OR "personal*" OR "custom*") AND ("virtual learn*" OR "virtual classroom" OR "vr") AND gami* AND ("learn* theory" OR "online collaborative learning" OR "engage*" OR "motivat*" theory)) **Date of Access: May 2024** |
| ScienceDirect | (("adaptive" OR "personalize" OR "customize") AND gamification AND ("virtual learning" OR "vr") AND ("learning theory" OR "online collaborative learning" OR "motivation theory")) **Date of Access: May 2024** |

through a systematic search of electronic databases, (2) screening of titles and abstracts to exclude irrelevant or duplicate records, (3) eligibility assessment of full-text articles based on predefined inclusion and exclusion criteria, and (4) inclusion of studies that met all criteria for detailed analysis.

## Identification

Several key steps in the systematic review process were used to choose a great deal of relevant literature for this study. First, keywords are selected, and then related terms are searched for using dictionaries, thesaurus, encyclopaedias, and past research. All relevant terms were selected after search strings for the Scopus, Web of Science, IEEE, and ScienceDirect databases were created (see Table 1). During the first stage of the systematic review process, 666 publications were successfully collected for the current study project from both databases.

## Screening

During the screening phase, the gathered research items are evaluated to determine their relevance to the predefined research questions. The content-related criteria often employed in this phase involve selecting studies pertinent to immersive virtual learning for technical skills training. Duplicates are removed from the list of identified articles at this stage. Initially, 454 publications were excluded, and the subsequent stage assessed 212 articles based on various inclusion and exclusion criteria specific to this study (refer to Table 2).

The primary criterion was research articles, as they provide the main source of practical recommendations. While the review initially considered a range of literature, including reviews, meta-syntheses, meta-analyses, book, book series, chapters, and conference proceedings, these sources were ultimately excluded due to concerns regarding their methodological rigor and scope. Conference proceedings often undergo a less stringent peer-review process and primarily present preliminary findings rather than

**Table 2 Inclusion and exclusion criteria.**

| Criterion | Inclusion | Exclusion |
|---|---|---|
| Language | English | Non-English |
| Timeline | 2019–2024 | <2019 |
| Literature type | Journal (Article) | Conference, Book, Review |

comprehensive, validated research. Similarly, book chapters may not always adhere to standardized reporting guidelines, limiting their reproducibility and comparability with journal articles. Given the systematic nature of this review, prioritizing peer-reviewed journal articles ensures a more rigorous, high-quality synthesis of research findings. The review was restricted to publications in English and focused on the period from 2019 to 2024. A total of 98 publications were rejected due to duplication.

## Eligibility

A total of 55 articles proceeded to the eligibility phase. At this stage, both the titles and full texts were assessed to ensure alignment with the study's aim. Following this assessment, twelve publications were excluded as their focus, based on title, abstracts, or core content, did not meet the inclusion criteria. This resulted in a final set of 43 articles included for synthesis. While we acknowledge that some conference proceedings particularly those from top-tier venues are subject to rigorous peer review and contribute valuable early-stage research, we opted to exclude them in this review. This decision was made to maintain consistency in methodological rigor, reporting depth, and completeness of data across studies. Conference articles often present condensed findings with limited methodological detail, making them less suitable for the type of in-depth synthesis required in this systematic review. That said, we recognize the value of high-quality conference work and suggest that future reviews could incorporate such studies by evaluating their inclusion based on venue quality or methodological robustness. A complete list of the included journal articles is provided in the Supplemental Materials to ensure transparency and replicability.

## Data abstraction and analysis

This study employed an integrative analysis approach to examine and synthesize diverse research designs, with a particular emphasis on quantitative studies. The objective was to identify key themes and subthemes related to adaptive gamification in collaborative virtual classrooms. The process began with data extraction from the 43 included studies, focusing on information relevant to the research questions, such as theoretical frameworks, gamification strategies, learner engagement outcomes, and collaborative learning components.

Thematic analysis was conducted using an inductive coding process. Initially, two researchers independently reviewed each article and performed open coding to identify recurring patterns, keywords, and concepts. These initial codes were then grouped into broader categories through axial coding to form preliminary themes. The coding process
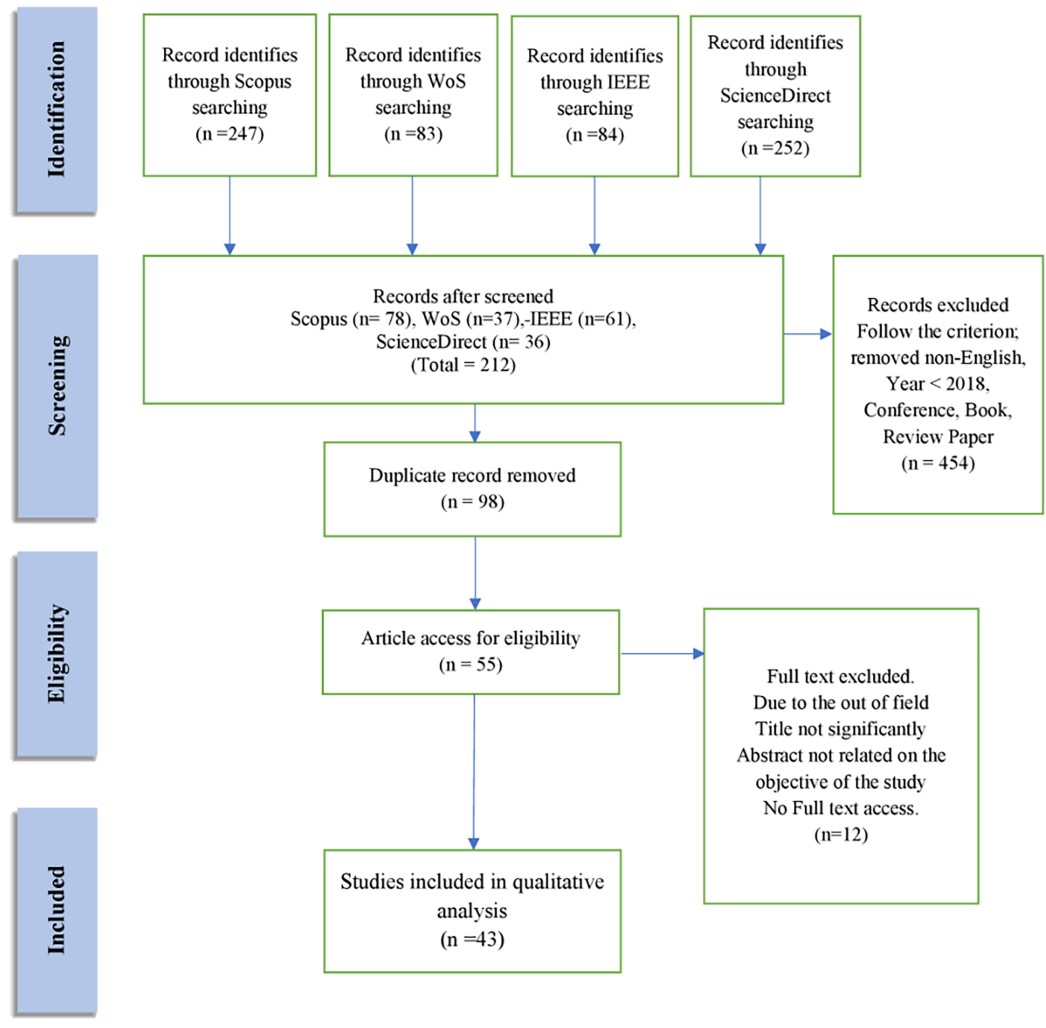

**Figure 1 Flow diagram of the proposed searching study.**

was facilitated using a shared spreadsheet matrix, where extracted data were mapped according to study characteristic, gamification features, learner outcomes, and theoretical underpinnings.

To ensure consistency and reliability, regular peer debriefing sessions were conducted throughout the coding process. Discrepancies in interpretation or thematic classification were resolved through discussion until consensus was reached. In cases where disagreements persisted, the third and fourth researchers was consulted to provide an independent opinion. A thematic logbook was maintained throughout the analysis to document coding decisions, emerging insights, and changes to theme definitions, ensuring auditability and transparency.

The finalized themes were reviewed holistically to identify conceptual overlaps or contradictions, which were then refined to enhance clarity and thematic coherence. Figure 1 illustrates the analytical process, showing how the 43 studies contributed to the development of major themes. This rigorous and collaborative approach to data

**Table 3 Framework/theory influence motivation and engagement in gamification for virtual learning.**

| No | Framework/Theory | References |
|---|---|---|
| 1 | Self-determination theory (SDT) | *Botte, Bakkes & Veltkamp (2020)*, *Botte et al. (2022)*, *Farikah et al. (2023)*, *Chukwu (2024)*, *Dumas Reyssier et al. (2023)*, *Kian, Sunar & Su (2022)* |
| 2 | Constructivism | *Yusoff & Shafiril (2019)*, *Xavier (2020)*, *Ros et al. (2020)*, *Velaora et al. (2022)*, *Abdirahma et al. (2023)*, *He, Ratanaolarn & Sitthiworachart (2024)*, *lbeigi, Bairaktarova & Ehsani (2024)* |
| 3 | Octalysis framework | *Kian, Sunar & Su (2022)* |
| 4 | ARCS model | *Jeong (2019)*, *Velaora et al. (2022)* |
| 5 | Flow theory | *Tramonti et al. (2021)*, *Chukwu (2024)* |
| 6 | Cognitive load theory | *Chukwu (2024)* |

abstraction and synthesis aimed to enhance the reliability, replicability, and validity of the review findings.

# RESULTS

This section reports the finding of 43 studies published on adaptive gamification in collaborative virtual classroom from 2019 to 2024, as systemized in the table below for each research questions.

## RQ1: how does gamification influence learner motivation and engagement in virtual learning environments?

This section analyze how gamification affects learner motivation and engagement, considering both adaptive and non-adaptive approaches. Articles here discuss motivational theories, learning theories, and long-term impacts of gamified environments. Table 3 shows framework or theories that influence learner motivation and engagement in virtual learning environment.

Studies shows that gamification can significantly boost student engagement and motivation in virtual learning environments when it aligns with established motivational theories. The SDT emerges as a fundamental framework, emphasizing key motivational elements such as competence, autonomy, relatedness, and personalization. *Botte et al. (2022)* demonstrated that reward systems designed around SDT principles effectively enhanced student motivation by adapting to individual learning needs. This adaptive approach allows the learning system to respond dynamically to student behavior, resulting in improved motivational outcomes. Similarly, *Farikah et al. (2023)* shows that incorporating SDT into e-learning environments support engagement, particularly in team-based learning contexts. When virtual learning environments address core psychological needs such as fostering a sense of competence and autonomy, students are more likely to remain engaged over longer periods. These findings align with earlier research by *Koivisto & Hamari (2019)*, which identified that the integration of game-like elements in online platforms serves as an effective motivator by creating engaging learning experiences that resonates with users' intrinsic desire for achievement and progress.

Operational SDT within adaptive gamification systems requires a structured approach to effectively foster learners' needs for competence, autonomy, and relatedness. To address

the need for competence, gamification systems should incorporate progressive challenges, timely feedback, and opportunities for mastery. AI-driven algorithms can dynamically monitor learner performance and adjust challenge levels, ensuring an optimal balance between engagement and difficulty (*Luarn, Chen & Chiu, 2023*). Implementing tiered recognition systems, such as badges or rankings (*e.g.*, beginner, intermediate, expert), can acknowledge skill development, while personalized feedback mechanisms can enhance motivation by highlighting strengths and suggesting areas for improvement (*Gupta & Goyal, 2022*; *Sailer & Homner, 2019*). To satisfy the need for autonomy, adaptive gamification systems should empower learners by giving them control over their educational experiences. This can be achieved through features such as branching narratives, open-ended quests, and self-directed learning modules, which allow students to take ownership of their progress (*Gupta & Goyal, 2022*). Gamified goal-setting tools and progress-tracking dashboards can further support autonomy by enabling learners to set and monitor personal learning goals. Additionally, customization options such as avatar selection, difficulty preferences, and individualized learning pathways ensure the system adapts to diverse motivation profiles, aligning with frameworks like the FSLSM and HEXAD player types. Finally, to foster a sense of relatedness, adaptive gamification systems should promote collaboration and a sense of belonging. Dynamic mechanisms can assign learners to teams based on skill levels, learning styles, or motivation profiles, ensuring balanced and productive collaboration (*Dindar, Ren & Järvenoja, 2021*). Social missions and cooperative challenges can encourage teamwork, while features like mentorship leaderboards and peer-support systems can recognize and reward students for helping their classmates. By integrating these elements, adaptive gamification systems can create a supportive and engaging learning environment that aligns with the principles of SDT.

In another study, *Dumas Reyssier et al. (2023)* explored the impact of adaptive gamification on motivation within secondary education. It was found that adaptive strategies were most effective after several lessons, where they could tailor the experience to the varying motivational levels of learners. Therefore, the study suggested that gamification elements need to be flexible and adaptive to maintain their effectiveness over time, especially when addressing diverse student needs. The incorporation of constructivist principle alongside gamification ensures that students are actively involved in learning process, which leads to deeper engagement and retention (*Velaora et al., 2022*). Likewise, *Xavier (2020)* demonstrated that gamified, quest-based activities significantly increased motivation among language learners, emphasizing the efficacy of structured challenges and rewards in maintaining student interest. Meanwhile, the combination of gamification with theories like Flow and Attention, Relevance, Confidence, and Satisfaction (ARCS) has also shown promising results in promoting engagement. For example, *Chukwu (2024)* found that aligning game elements with course objectives using SDT and Flow Theory led to higher levels of students' motivation and learning outcomes. By balancing cognitive challenges with appropriate rewards, the system can maintain a state of flow, where learners remain deeply engaged without becoming overwhelmed or bored.

**Table 4 Theory/model to personalize game elements.**

| No | Model/Theory | References |
|---|---|---|
| 1 | Felder-Silverman learning style model (FSLSM) | *El-Bishouty et al. (2019)*, *Aljabali et al. (2020)*, *Altaie & Jawawi (2021)*, *Bennis, Kandali & Bennis (2022)*, *Kang & Kusuma (2020)*, *Rodrigues et al. (2024)*, *Scott & Campo (2023)*, *Zaric et al. (2021)* |
| 2 | HEXAD player type model | *Hallifax et al. (2019b)*, *Lopez & Tucker (2021)*, *Rodríguez, Puig & Rodríguez (2021)* |
| 3 | Myers-Briggs type indicator (MBTI) model | *Fatahi (2019)*, *Leclercq et al. (2020)* |
| 4 | Big five personality | *Bhalerao et al. (2021)*, *Hallifax et al. (2019b)*, *Kang & Kusuma (2020)* |
| 5 | Multiple intelligence theory | *Mohamad et al. (2019)* |

The reviewed studies emphasize that strategically implementing gamification, grounded in theories such as SDT, Flow, and constructivism, significantly enhances motivation and engagement in virtual learning environments. Future research should refine adaptive gamification strategies to optimize personalization and long-term learner engagement. Additionally, evaluating the effectiveness of these interventions requires robust measurement frameworks that assess competence, autonomy, and relatedness. Performance analytics, including test scores, skill progression, and completion rates, can provide insights into competence development. Engagement tracking, such as learners' interaction with adaptive pathways, self-paced progression, and goal-setting dashboards, can measure autonomy. Relatedness should be examined through collaborative participation metrics, social interactions, and peer-supported learning activities. Behavioral indicators like participation trends, dropout rates, and sustained motivation levels can further validate the success of adaptive gamification models. Future studies should integrate longitudinal assessments to analyze how gamified strategies impact knowledge retention, motivation sustainability, and overall learning effectiveness in diverse educational settings.

## RQ2: how can gamification be tailored to individual to enhance engagement and motivation in virtual collaborative classroom?

This section explores how gamification elements can be adapted to accommodate diverse learner profiles, with a focus on how such personalization enhances individual learning experiences within collaborative virtual environments. A summary of the models and theories used to guide these adaptations is provided in Table 4.

Recent research into adaptive gamification highlights its potential to enhance engagement and motivation by aligning game-based strategies with individual learner characteristics. A range of theoretical frameworks have been employed to inform personalized gamification, including the FSLSM, HEXAD Player Type model, Myers Briggs Type Indicator (MBTI), Big Five Personality model, and Multiple Intelligence Theory.

Among these, FSLSM has been widely applied to adapt gamification design based on cognitive and perceptual learning preferences. The model classifies learners across four

dimensions—active/reflective, sensing/intuitive, visual/verbal, and sequential/global—offering a structured basis for tailoring content delivery, interaction styles, and motivational cues. In adaptive gamified environments, FSLSM has been used to inform real-time personalization through learner profiling, dynamic game mechanics, customized feedback, and AI-driven adaptation. For instance, *Altaie & Jawawi (2021)* found that FSLSM-based gamification significantly improved motivation and performance among children in computational thinking tasks. *El-Bishouty et al. (2019)* also successfully integrated FSLSM into learning management systems to boost student engagement, while *Bennis, Kandali & Bennis (2022)* demonstrated that FSLSM-informed game design improved learning outcomes by aligning gameplay with individual learning preferences.

While FSLSM offers strong instructional alignment, alternative frameworks also provide meaningful pathways for personalization particularly in contexts where motivation, behavior, or learner identity is central. For example, Multiple Intelligence Theory has been applied to support critical thinking among technical and vocational students *Mohamad et al. (2019)* and *Fatahi (2019)* demonstrated the effectiveness of combining MBTI with emotion modelling (OCC) to personalize learning interaction. Meanwhile, *Bhalerao et al. (2021)* utilized the OCEAN model (Big Five) to create gamified tools for career guidance, effectively supporting adolescents in self-directed exploration.

Other than that, the HEXAD Player Type model offers a complementary lens by classifying learners into six motivational archetypes: Achiever, Socializer, Free Spirit, Philanthropist, Player, and Disruptor, where each associated with different responses to gamification mechanics. Research shows that matching game elements to these player types can significantly enhance motivation and engagement (*Hallifax et al., 2019b*; *Lopez & Tucker, 2021*; *Rodríguez, Puig & Rodríguez, 2021*). Taking integration further, *Kang & Kusuma (2020)* created a hybrid approach combining FSLSM with the Big Five Personality model. Their research in foreign language learning demonstrated that this combined approach enhanced both learning outcomes and student motivation.

Collectively, these studies underscore that no single model is universally optimal; rather, their effectiveness is context-dependent. FSLSM is particularly well-suited for instructional personalization, while personality-based models like HEXAD, MBTI, and Big Five are often more effective in addressing motivation, engagement style, and learner identity. The success of hybrid approaches suggests that multi-dimensional personalization incorporating both cognitive and affective factors offers a promising direction for adaptive gamification design.

In conclusion, empirical evidence across these frameworks supports the practical value of adaptive gamification. When game elements are thoughtfully aligned with learners' preferences, personality traits, and learning styles, students exhibit increased motivation, deeper engagement, and improved academic performance. These findings reinforce the importance of integrating personalization mechanisms into educational gamification systems. Future work should continue exploring longitudinal impacts, the interplay between different models, and the potential of AI-driven personalization to deliver flexible, learner-centered virtual learning environments.

**Table 5 Online learning theory/model.**

| No | Theory/Model/Framework | References |
|---|---|---|
| 1 | Online collaborative learning theory (OCL) | *Saçak & Kavun (2020)*, *Torres et al. (2021)*, *Hao & Tasir (2024)* |
| 2 | Community of inquiry (CoI) | *Timonen & Ruokamo (2021)* |
| 3 | Social learning theory | *Nyembe & Howard (2019)* |
| 4 | Multimodal interaction theory | *Doumanis et al. (2019)* |
| 5 | 5E learning cycle model | *Liu & Lu (2021)* |
| 6 | Gamified collaborative | *Hasan, Nat & Vanduhe (2019)* |
| 7 | Cognitive load theory (CLT) | *Chukwu (2024)* |
| 8 | PADDIE M+ Model | *Páez-Quinde et al. (2023)* |
| 9 | Cooperative learning theory | *Rudolf (2022)* |
| 10 | Goal-access-feedback-challenge-collaboration (GAFCC) model | *Bai et al. (2022)* |

## RQ3: how collaborative and engagement among learners in virtual classrooms can be enhance with gamification?

This section explores online learning theories that encourage social learning for collaboration and teamwork among learners. It also evaluates the influence of these theories in gamification system on interaction quality and group cohesion in virtual settings. Results can be referred to Table 5.

Findings reveals that incorporating gamification with established online learning theories can substantially enhance collaboration and engagement among learners in virtual classroom settings. There are several theories, models and frameworks that can be adopted in this context. Online Collaborative Learning theory (OCL) proposed by *Harasim (2015)* promotes knowledge construction through interaction and social engagement, which can further be strengthened by gamified elements. Adding features such as group challenges and rewards to the OCL framework to incentivizes students to actively engage in collaborative activities, work toward shared objectives, and maintain sustained participation (*Hao & Tasir, 2024*; *Saçak & Kavun, 2020*; *Torres et al., 2021*). Similarly, the Community of Inquiry (CoI) framework suggests that fostering a balance among cognitive, social, and teaching presence is critical in virtual classrooms (*Timonen & Ruokamo, 2021*). Gamification elements such as leaderboards and challenges can enhance social presence by encouraging students to actively participate in discussions and collaborative activities.

Other than that, a significant finding emerges from the integration of Social Learning Theory (*Nyembe & Howard, 2019*) with the GAFCC model (*Bai et al., 2022*), suggesting that goal-oriented collaborative activities, when gamified, can create more engaging learning experiences. The GAFCC model's emphasis on feedback and challenges aligns with the cognitive engagement principles outlined in Cognitive Load Theory (CLT) (*Sweller, 1988*). In a recent study, *Chukwu (2024)* applied these principles to demonstrate the well-designed gamification elements must balance engagement with cognitive processing capabilities. Besides, The 5E Learning Cycle model provides a structured approach to implementing gamified collaborative activities (*Liu & Lu, 2021*), while the PADDIE M+ model (*Páez-Quinde et al., 2023*) offers a systematic framework for designing

and implementing these interventions. These models, when considered alongside Multimodal Interaction Theory (*Doumanis et al., 2019*), suggest that successful gamification in virtual classrooms should incorporate multiple modes of interaction to support diverse learning styles and enhance engagement. Research by *Rudolf (2022)* that implemented the Collaborative Learning Theory further reinforces the importance of structured interdependence in gamified collaborative activities. This theoretical perspective, when combined with the principles of gamification, indicates that collaborative tasks should be designed to promote positive interdependence while maintaining individual accountability through game mechanics.

Despite these promising findings, several open challenges remain in the field of adaptive gamification. First, scalability remains a critical issue, as many adaptive systems struggle to maintain effectiveness across larger and more diverse learner populations. Future research should explore scalable frameworks that can dynamically adjust to the needs of a broader audience without compromising personalization. Second, existing frameworks often lack robust mechanisms for long-term engagement, as motivational levels can fluctuate over time. Researchers should investigate strategies to sustain engagement beyond short-term interventions, such as incorporating real-time analytics and AI-driven adaptation. Third, there is a need for more comprehensive studies on the limitations of current adaptive systems, particularly in addressing cultural, cognitive, and contextual differences among learners. Finally, the integration of emerging technologies, such as machine learning and immersive environments, could offer new avenues for enhancing adaptive gamification systems. By addressing these challenges, researchers can develop more effective and inclusive frameworks that maximize the potential of adaptive gamification in education. Besides, the findings also collectively suggest the effective enhancement of collaboration and engagement in virtual classrooms through gamification requires a multi-theoretical approach that considers social, cognitive, and pedagogical dimensions. The synthesis of these frameworks provides a robust foundation for designing adaptive gamification strategies that can effectively promote meaningful collaboration while maintaining high levels of learner engagement in virtual classroom environments.

## DISCUSSION

Based on the findings derived from the thematic analysis, this study proposes a theoretical framework (Fig. 2) that conceptualizes the relationship between adaptive gamification and its impact on learner motivation, personalization, and collaboration within virtual classroom settings. This framework integrates key constructs identified across the reviewed literature, including Self-Determination Theory, constructivist learning principles, learning style alignment, and collaborative learning models. It serves as a guiding structure for understanding how adaptive gamification mechanisms can be designed to address diverse learner needs while fostering sustained engagement and effective group interaction. The following discussion is organized around this framework's three central dimensions, each corresponding to one of the research questions: (1) motivation and engagement,

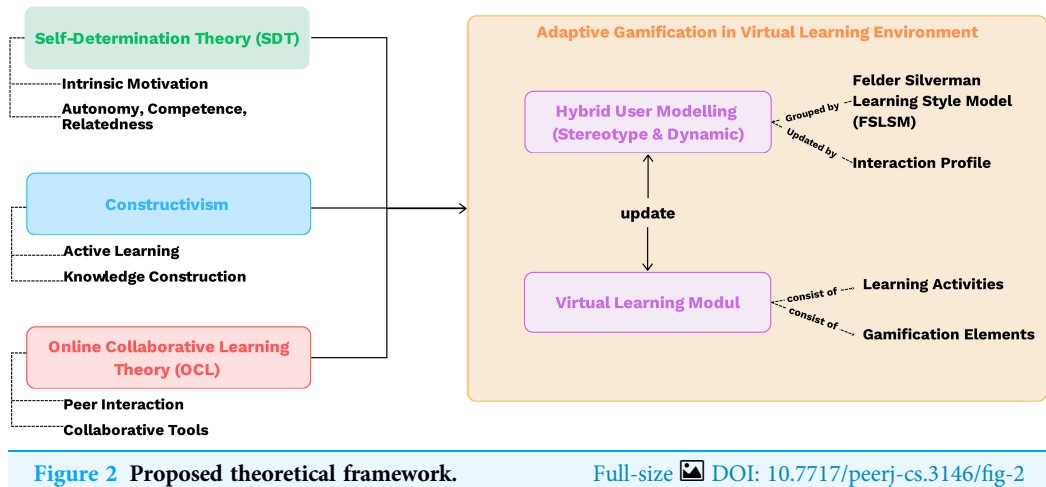

Figure 2 **Proposed theoretical framework.**

(2) adaptive personalization through learning styles, and (3) collaborative learning enhancement.

## Subtheme 1: motivation and engagement through gamification in virtual learning

This subtheme explores how gamification elements such as points, badges, leaderboards, and challenges affect learner motivation and engagement. The findings suggest that well-designed gamified environments can sustain interest and participation over time, especially when informed by established theories such as SDT and constructivism.

Studies reviewed indicate that aligning gamified features with SDT principles which are autonomy, competence, and relatedness significantly enhance learner motivation (*Botte et al., 2022*; *Farikah et al., 2023*). When these psychological needs are met, learners are more intrinsically motivated and show improved academic outcomes. The versatility of SDT across educational settings supports its role in guiding adaptive gamification that addresses diverse learner profiles. In virtual contexts, where social disconnection can reduce engagement, fostering relatedness through collaborative game-based tasks becomes particularly valuable. For example, *Jeong (2019)* found that such activities promote peer interaction and deeper learning. Furthermore, the dynamic responsiveness of SDT-based gamification able to adjust to learners' changing motivations has been shown to be essential in virtual learning environments, where engagement is often unstable (*Dumas Reyssier et al., 2023*).

Complementing SDT, constructivism enhances gamification design by emphasizing the learner's active role in building knowledge through meaningful interaction (*Velaora et al., 2022*; *Xavier, 2020*). Game elements that promote exploration, critical thinking, and problem-solving not only increase engagement but also support the development of higher-order cognitive skills. *Velaora et al. (2022)* demonstrated that gamified constructivist activities in digital design education boosted student creativity and analytical

abilities. This evolution reflects a shift in gamification practices—from surface-level incentives to cognitively enriching experiences.

Together, SDT and constructivism provide a synergistic foundation for adaptive gamified systems. While SDT supports motivation through psychological needs, constructivism ensures relevance and engagement through learner-driven activity. This dual approach is especially effective in virtual learning, where passive formats often lead to disengagement. Unlike static models, adaptive systems based on these theories can continuously evolve to meet changing learner needs. This theoretical pairing offers a sustainable strategy for maintaining long-term motivation and academic success in online environments.

However, challenges remain. There is a risk that gamified features may be superficially aligned with SDT or constructivist principles without genuinely supporting learner engagement or knowledge construction. Future studies should assess not only the presence of theoretically grounded elements but also their pedagogical coherence and authenticity from the learner's perspective. Longitudinal research is also needed to examine whether motivational and cognitive benefits persist, particularly in self-paced or asynchronous learning contexts where intrinsic motivation is crucial. Moreover, incorporating real-time data such as behavioral patterns, affective feedback, and performance metrics, could significantly improve the precision and responsiveness of adaptive gamified interventions.

While motivating learners is central to sustaining participation in virtual classrooms, it is equally important to recognize that motivation alone cannot fully address diverse cognitive needs. Effective engagement strategies must account for how learners process information and interact with content. Personalization, therefore, becomes the logical progression in refining adaptive gamification. The next subtheme builds on this foundation by exploring how tailoring gamified experiences to individual learning styles particularly through the FSLSM can optimize both motivation and cognitive engagement. By aligning game elements with learners' preferred modes of learning, adaptive systems can support deeper, more meaningful learning pathways.

## Subtheme 2: adaptive gamification and learning style

This subtheme explores the role of personalized gamification in enhancing engagement and motivation by aligning design with individual learning preferences. Among various frameworks used in adaptive gamification, the FSLSM features prominently due to its structured approach to identifying learners' cognitive and perceptual styles.

FSLSM categorizes preferences across four dimensions: active/reflective, sensing/intuitive, visual/verbal, and sequential/global, offering a detailed foundation for tailoring instructional strategies and gamified interactions. When applied to virtual learning environments, FSLSM has been associated with improvements in learner motivation, satisfaction, and academic outcomes. For instance, research by *Altaie & Jawawi (2021)* reported enhanced computational thinking skills in young learners using FSLSM-based gamification. Similarly, *El-Bishouty et al. (2019)* and *Bennis, Kandali & Bennis (2022)* demonstrated that adapting content delivery to FSLSM dimensions supports increased engagement and participation. FSLSM thus supports the creation of virtual learning

environments that dynamically adapt to learners' evolving profiles, promoting both cognitive and emotional engagement.

However, while FSLSM has shown promise, it should not be viewed as universally superior to other personalization frameworks. Alternative models such as the HEXAD Player Type model, the Myers-Briggs Type Indicator (MBTI), and the Big Five Personality Traits offer distinct advantages, particularly in contexts where motivational orientation or personality-driven engagement is central to the learning design. HEXAD, for example, classifies learners into six user types such as Achiever, Socializer, or Free Spirit, where each responding differently to game elements like competition, exploration, or altruism (*Hallifax et al., 2019b*; *Rodríguez, Puig & Rodríguez, 2021*). These models provide complementary insights that may be more suitable in designing motivational strategies or social interaction mechanics, rather than instructional sequencing.

By contrast, FSLSM is more directly concerned with how learners process and engage with educational content, offering a practical and pedagogically grounded basis for content personalization. This specificity is especially valuable in instructional contexts where fine-tuned adaptation is essential. Empirical findings by *Aljabali et al. (2020)* and *Bennis, Kandali & Bennis (2022)* support this distinction, showing that FSLSM-based personalization improves learning outcomes and satisfaction more consistently than broader trait-based models. Besides, hybrid approaches are increasingly explored to leverage the strengths of multiple frameworks. *Kang & Kusuma (2020)*, for instance, combined FSLSM with the Big Five Personality model in language learning and found that integrated personalization enhanced both engagement and academic performance. Such evidence suggests that the effectiveness of a personalization model is often context-dependent and may benefit from multi-dimensional designs.

Nevertheless, the application of FSLSM and similar learning-style models continues to attract criticism. Some scholars question the empirical validity of matching instruction to preferred styles and argue that such alignment may limit opportunities for learners to develop cognitive flexibility (*Alalouch, 2021*; *Husmann & O'Loughlin, 2018*). To mitigate these concerns, future research should explore adaptive gamification systems that go beyond static categorization. Incorporating real-time learner analytics, emotional cues, and performance data could enable systems to respond dynamically and developmentally to evolving learner needs.

In summary, FSLSM offers a robust foundation for tailoring gamified learning based on cognitive processing preferences, but it is not a one-size-fits-all solution. Models like HEXAD, MBTI, and Big Five contribute valuable dimensions, particularly in addressing user motivation, personality, and engagement style. The choice of framework should align with the specific goals, context, and learner profile of a given educational setting. A balanced, evidence-based approach that considers hybrid or dynamic personalization strategies will be key to advancing adaptive gamification systems that are both pedagogically sound and learner-centered.

While tailoring learning experiences to individual preferences enhances motivation, it is equally important to recognize the collaborative nature of knowledge construction. Learning is not solely a cognitive task but a social process enriched by interaction, dialogue,

and shared problem-solving. Therefore, beyond individual adaptation, effective adaptive gamification should also foster meaningful peer engagement. The next subtheme explores how gamification strategies guided by OCL theory can promote teamwork, social presence, and higher-order thinking in virtual classrooms. This focus highlights the essential role of collaboration in sustaining learner motivation and deepening cognitive engagement.

## Subtheme 3: gamification and collaborative learning in virtual classroom

This subtheme explores how gamification enhances collaboration among learners, particularly through the lens of OCL theory. Game-based mechanics such as team quest, peer feedback, and shared goals support interaction, cooperation, and collective problem-solving, ultimately enriching the virtual learning experience through social engagement. OCL provides a strong theoretical foundation for designing digital environments that promote structured collaboration and community-driven knowledge construction. Across the literature, OCL-informed gamification has consistently demonstrated positive impacts on collaborative learning. For example, *Saçak & Kavun (2020)* found that using gamified tools like Flipgrid increased active participation and deeper engagement. Similarly, *Torres et al. (2021)* highlighted improvements in learner satisfaction and usability when gamification was intentionally designed to support collaboration. These study affirm that OCL, when embedded within adaptive gamified system, fosters interactive and engaging environments that promote meaningful peer learning.

The social dimension of learning is central to OCL's relevance. By emphasizing peer interaction and communal knowledge-building, OCL aligns closely with gamified strategies that promote shared learning goals (*Nyembe & Howard, 2019*). This approach positions adaptive gamification not only as a tool for individual engagement but also as a vehicle for cultivating collaborative, learner-driven communities. In virtual classrooms, where engagement is often shaped by social dynamics, OCL-based designs can deepen learner interaction and enhance motivation. Furthermore, OCL-integrated gamification has been linked to the development of higher-order thinking skills. *Hao & Tasir (2024)* demonstrated that collaborative tasks in gamified MOOCs guided by OCL principles support critical thinking and problem-solving. This alignment with Bloom's Taxonomy indicates that gamification can scaffold both social and cognitive development, creating intellectually rigorous virtual learning environments.

OCL's adaptability across diverse contexts is another key strength. Research by *Doumanis et al. (2019)* and *Páez-Quinde et al. (2023)* underscore underscores the model's flexibility in accommodating varied learner needs, preferences, and group collaboration styles. Its compatibility with synchronous and asynchronous learning modes further supports its integration into adaptive systems, as evidenced by studies from *Timonen & Ruokamo (2021)* and *Torres et al. (2021)*. Whether in live discussions, asynchronous projects, or blended environments, OCL-informed gamification can be customized to support different instructional formats. In addition, the motivational dimension of gamified collaboration is well-supported in the literature. *Chukwu (2024)* identified the positive influence of gamified systems on learner motivation and

performance, while *Bai et al. (2022)* demonstrated how elements like fantasy significantly improved engagement and collaboration. These findings suggest that when grounded in OCL, gamification can foster both sustained participation and team cohesion.

However, the design of such systems must be carefully managed. While competition and point-based rewards can increase motivation, they also risk undermining collaboration if not thoughtfully implemented. Overemphasis on individual achievement may foster excessive competitiveness, weakening group cohesion. To mitigate this, researchers have advocated for mechanisms such as adaptive role assignments, shared reward systems, and collaborative quests that balance individual accountability with collective success.

In conclusion, OCL provides a robust framework for enhancing collaboration in adaptive gamified virtual learning environments. It supports not only engagement and teamwork but also the development of cognitive skills through socially grounded learning designs. Future research should focus on refining gamified elements such as adaptive learning pathways, dynamic role assignment, and context-sensitive feedback to optimize collaboration without compromising educational goals. Longitudinal studies across disciplines, learner profiles, and cultural settings will be essential for validating the long-term effectiveness and scalability of OCL-based gamification. Additionally, integrating emerging technologies such as AI and learning analytics can offer real-time, personalized support for collaborative processes, advancing the precision and responsiveness of these systems.

Together, the three themes, motivation and engagement, adaptive personalization through learning styles, and collaborative learning present a comprehensive framework for adaptive gamification in virtual education. By integrating SDT and constructivism, educators can address intrinsic motivation and cognitive engagement. The use of FSLSM offers a systematic approach to personalization, while OCL ensures that learning remains socially grounded and interactive. This multidimensional model transcends traditional gamification by fostering inclusive, dynamic, and learner-centered environments that respond to both individual and group learning needs.

### Study limitations

Despite the contributions of this review, several limitations must be acknowledged. First, the scope of the included studies was limited to peer-reviewed journal articles, which excluded other potentially valuable literature such as conference proceedings, book chapters, and grey literature. While this decision was made to ensure consistency in methodological rigor and reporting standards, it may have resulted in the omission of early-stage research or novel frameworks often presented in non-journal sources. We recognize that some high-quality conference proceedings can offer significant insights, and future reviews could consider including them using venue rankings or methodological appraisal criteria.

Second, most of the reviewed studies were limited in scope and duration, which raises concerns about the generalizability and long-term applicability of the synthesized findings. The variability in study contexts, learner demographics, and instructional environments

further constrains the extent to which these results can be applied across diverse educational settings. Third, while this study proposes an integrated framework grounded in established motivational, learning, and collaborative theories, it remains conceptual. Empirical validation is necessary to determine its practical effectiveness. Future research should focus on validating this framework through real-world implementations, ideally using longitudinal study designs to examine sustained learner engagement, academic performance, and the adaptability of the model over time.

In addition, mixed-method approaches should be employed to capture the complex interplay between personalization, collaboration, and learner experience. Comparative analyses of different personalization models (*e.g.*, FSLSM *vs.* HEXAD or MBTI) may also yield further insights into optimizing adaptive gamification strategies. Lastly, ethical considerations related to adaptive data collection and learner profiling such as privacy, consent, and algorithmic transparency should be critically addressed to ensure the responsible and equitable use of adaptive technologies in educational contexts.

## CONCLUSIONS

This systematic review of adaptive gamification in collaborative virtual classrooms underscores its transformative potential in enhancing learner motivation, engagement, and peer collaboration through personalized, interactive strategies. Grounded in established motivational and learning theories such as SDT, constructivism, and the FSLSM, adaptive gamification has demonstrated its ability to align learning activities with individual learner preferences and styles, thus promoting sustained participation and improved learning outcomes. Furthermore, integrating collaborative learning principles within gamified environments has shown to support meaningful peer interaction, collective problem-solving, and mutual accountability are the key components of effective virtual classrooms.

However, several critical gaps warrant further exploration. Current evidence on the long-term effectiveness of adaptive gamification across varied subject domains and educational levels remains limited. Moreover, the ability of such systems to dynamically adapt in real time to learners' changing behaviours and preferences is still under-investigated. The influence of collaborative gamification strategies on group dynamics, knowledge retention, and equitable participation also requires deeper examination. Technological scalability and the practicality of implementing fully adaptive systems in diverse learning environments are other key challenges yet to be resolved.

Future research should move beyond theoretical modelling by conducting empirical evaluations of adaptive gamification frameworks across diverse educational contexts, including longitudinal studies that track learner engagement and outcomes over time. Mixed-method approaches involving both quantitative metrics and qualitative learner feedback will be especially valuable in uncovering nuanced insights into learner experiences, motivation trajectories, and collaborative behaviours. These directions will help educators and researchers co-create more dynamic, inclusive, and scalable gamified

virtual learning environments that respond effectively to the evolving needs of 21st-century learners.

## ACKNOWLEDGEMENTS

We used ChatGPT (OpenAI) to assist with grammar and sentence structure correction during the writing process. The tool was used solely to refine the clarity and fluency of the language, without altering the original content or meaning.

### Funding
This work was supported by a University Research Grant (GUP-2023-0074), and Faculty of Information Science and Technology (FTM-1) from Universiti Kebangsaan Malaysia. The funders had no role in study design, data collection and analysis, decision to publish, or preparation of the manuscript.

### Grant Disclosures
The following grant information was disclosed by the authors:
University Research Grant: GUP-2023-0074.
Faculty of Information Science and Technology (FTM-1), Universiti Kebangsaan Malaysia.

### Competing Interests
The authors declare that they have no competing interests.

### Author Contributions
- Intan Yusrina Zairon conceived and designed the experiments, performed the experiments, analyzed the data, performed the computation work, prepared figures and/or tables, authored or reviewed drafts of the article, and approved the final draft.
- Tengku Siti Meriam Tengku Wook conceived and designed the experiments, analyzed the data, performed the computation work, authored or reviewed drafts of the article, and approved the final draft.
- Syahanim M. Salleh analyzed the data, authored or reviewed drafts of the article, and approved the final draft.
- Hadi Affendy Dahlan analyzed the data, authored or reviewed drafts of the article, and approved the final draft.

### Data Availability
This literature review does not include any raw data.

### Supplemental Information
Supplemental information for this article can be found online at http://dx.doi.org/10.7717/peerj-cs.3146#supplemental-information.

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
