# Peer review of "Adaptive gamification in collaborative virtual classroom: a systematic review"

_PeerJ Computer Science, doi:10.7717/peerj-cs.3146_

## Round 0.1 · original submission · Major Revisions

Experts have now judged your manuscript and as you can read, their comments require a major revision. In particular, the reviewers require to better introduce the concept of adaptive gamification, including addressing highly related work, and to identify more actionable conclusions from the results. Please carefully revise your manuscript by taking into account all comments from the reviewers.

Reviewer 1 ·

Basic reporting

While the manuscript is written in professional English, the section on adaptive gamification lacks clarity in introducing the concept. The theoretical underpinnings are mentioned, but the connection between theories and practical application is vague. This diminishes the paper's utility for researchers seeking to build upon this framework.

The manuscript provides a foundational background and references relevant literature in adaptive gamification and virtual learning. However, significant references that address critical gaps in gamification design are missing. For instance:
- Gamidoc: The importance of designing gamification in a proper way (Bassanelli, Bucchiarone, Gini) provides insights into robust gamification design principles that could have enhanced the discussion.
- Lost in gamification design: A scientometric analysis (Bassanelli et al.) offers a comprehensive scientometric perspective that could contextualize the challenges in designing adaptive systems.
The omission of these references limits the depth of the literature review and weakens the discussion on the importance of thoughtful gamification design.

The manuscript's structure aligns with academic norms, but the depth and utility of the results fall short. While the figures and tables provide an overview, they lack nuanced insights that could guide future research. For example, Table 5 does not adequately address the specific challenges of integrating online learning theories into adaptive gamification systems.

The topic holds promise for multidisciplinary audiences; however, the presentation of adaptive gamification as a concept is insufficiently detailed. The manuscript would benefit from a clearer explanation of what adaptive gamification entails and how it differs from traditional gamification in practice.

Although the study focuses on recent literature (2019–2024), it fails to present results that are both informative and actionable. The results section synthesizes existing studies but does not provide meaningful insights or recommendations for researchers looking to address gaps in adaptive gamification. Open challenges, such as the scalability of adaptive systems or limitations in existing frameworks, are not sufficiently highlighted.

The introduction effectively identifies engagement and collaboration challenges in virtual classrooms but does not adequately define adaptive gamification. A more precise explanation of its key components, challenges, and distinctions from traditional gamification would help establish a clearer foundation for the study.

The results lack depth and practical relevance. While the theoretical frameworks are mentioned, their integration into actionable strategies is unclear. For instance:
- How can specific models like FSLSM or SDT be operationalized within adaptive gamification systems?
- What are the measurable outcomes for learners in collaborative virtual classrooms?
These gaps hinder the manuscript's utility for researchers seeking to develop or test adaptive gamification systems.

Strengths:
- Rigorous methodology using PRISMA.
- Inclusion of motivational and learning theories.

Limitations*:
- Adaptive gamification is not introduced clearly.
- Results are not sufficiently deep or informative for researchers in the field.
- Missing references and insufficient focus on open challenges reduce the paper's overall impact.

Recommendations for Improvement:
Clarify Adaptive Gamification: Provide a detailed explanation of adaptive gamification, including its design principles, challenges, and applications.
Enhance Results: Present deeper insights and actionable recommendations, particularly for researchers designing adaptive systems. Highlight open challenges, such as scalability, personalization, and integration with diverse learning models.
Include Missing References: Incorporate key works like:
Gamidoc: The importance of designing gamification in a proper way (Bassanelli, Bucchiarone, Gini).
Lost in gamification design: A scientometric analysis (Bassanelli et al.).
These references would provide valuable perspectives on the importance of systematic and effective gamification design.

Present Open Challenges: Explicitly outline unresolved issues in adaptive gamification, such as technical barriers, learner diversity, and ethical considerations.
Improve Practical Relevance: Include case studies or specific examples that showcase how adaptive gamification has been implemented successfully and the lessons learned.

Experimental design

The article content aligns with the journal's focus on educational technology and systematic reviews. It addresses a relevant and timely topic—adaptive gamification in collaborative virtual classrooms—and provides a systematic review aimed at synthesizing existing knowledge in this field. However, the article could better meet the journal's expectations by providing clearer definitions of adaptive gamification and its unique aspects compared to traditional gamification.

The authors employ a systematic review methodology following the PRISMA framework, which ensures transparency and replicability in study selection and analysis. Ethical considerations appear to have been adhered to, particularly in the systematic inclusion and exclusion of studies. However, the depth of the investigation is limited, as the results do not provide sufficient practical insights or actionable recommendations for researchers.

The methodology is adequately described, with clear explanations of the database searches, inclusion/exclusion criteria, and thematic analysis. The use of PRISMA ensures a structured approach, and the workflow is illustrated effectively in the figures. However, additional detail on how themes and insights were derived from the selected studies would strengthen the replicability of the review.

While the survey methodology appears systematic and comprehensive in scope, there are gaps in coverage. Critical references that address adaptive gamification design (e.g., Gamidoc: The importance of designing gamification in a proper way and Lost in gamification design: A scientometric analysis) are absent. Their inclusion could enhance the comprehensiveness and balance of the review. Additionally, the discussion lacks a critical perspective on the limitations and challenges of adaptive gamification frameworks.

Most sources are cited appropriately and paraphrased correctly, aligning with academic standards. However, the review would benefit from the inclusion of more diverse and recent references to provide a balanced and up-to-date perspective. Missing citations, particularly those addressing gamification design challenges, weaken the review's breadth and depth.

Yes, the review is logically organized into clear subsections, corresponding to the research questions and themes identified. The structure follows a logical flow from the introduction to methodology, results, and discussion. However, some subsections, particularly the results, could be enhanced with deeper analyses and clearer connections between theories and practical applications. This would improve the overall coherence and utility of the review.

Validity of the findings

The manuscript does not adequately address the impact or novelty of adaptive gamification as a concept, particularly in the context of collaborative virtual classrooms. While the rationale for the study is rooted in addressing gaps in engagement and collaboration, the benefit to the literature could be better articulated by emphasizing how the findings advance existing knowledge or address known challenges. The inclusion of more actionable insights would also make the review more replicable and beneficial for future research.

The conclusions are generally consistent with the original research questions, summarizing the findings related to motivational theories, learning styles, and collaborative learning frameworks. However, the conclusions are not sufficiently detailed or critical. They do not delve deeply into the practical implications or address the limitations of the current findings. This weakens their connection to the study’s stated objectives and diminishes their utility for researchers seeking actionable guidance.

The argument presented in the review partially meets the goals set out in the introduction. The authors identify a clear objective of synthesizing theoretical and practical insights into adaptive gamification. However, the argument lacks depth and specificity, particularly in explaining how adaptive gamification can be effectively implemented in collaborative virtual classrooms. The discussion is more descriptive than analytical, leaving key aspects of adaptive gamification underexplored.

The conclusion acknowledges some gaps, such as the need for adaptive gamification strategies that address diverse learner needs. However, it falls short in specifying unresolved questions or suggesting concrete future research directions. For example:
- How can adaptive gamification frameworks be operationalized for different learning contexts?
- What are the technical and pedagogical challenges in implementing such systems?
- How can long-term impacts of adaptive gamification on learner outcomes be assessed?
Providing detailed future directions would greatly enhance the paper's value to researchers in this field.

Additional comments

Recommendations for Improvement:
1. Clearly articulate the novelty and potential impact of the findings within the broader literature.
2. Provide more specific conclusions that align with the research questions and include actionable recommendations.
3. Develop a more critical and analytical argument to strengthen the connection between the study's objectives and findings.
4. Identify specific unresolved questions and propose detailed future directions to guide subsequent research efforts.

Cite this review as

Reviewer 2 ·

Basic reporting

1. The paper is well written with some minor errors:
- Please ensure that each acronym is introduced and defined only once, at its first occurrence in the text.
- Line 283: "One of them are"
- The reference on line 297 appears to create confusion by suggesting it refers to Cognitive Load Theory (CLT) when, in fact, it pertains to one of the included studies. Ensure that the reference is explicitly connected to the study rather than the theory.
- Line 305 "Cooperative Learning Theory" is duplicated

2. The introduction clearly presents the context, the motivation and the goal of the study.

3. Findings are organised into themes, aiding readability. However, to avoid redundancy with the answers to the research questions, overlapping content should be merged and the relationship between subthemes and the research questions (how subthemes relate to or build upon the research questions) should be clearly explained

4. Not all readers will be familiar with the theories and models (e.g., Self-Determination Theory, Constructivism, Felder-Silverman Learning Style Model). A background section would make the paper more accessible to a broader audience.
Authors could include a concise overview of theories and models like SDT, FSLSM, and OCL and group theories by themes (e.g., motivation, learning styles, collaboration).
I would position this section before the methodology to provide context.

5. References (both the included studies and the additional references) are presented in a single list. However, having two separate lists would enhance transparency (i.e., clearly indicating which studies form the basis of the review's findings), ease of access (i.e., helping readers quickly locate primary sources and assess their relevance, and readability (i.e., avoiding confusion with background or supporting literature).
I would highlight studies included in the review differently (e.g., in a separate table) and group references into "Included Studies" and "Additional References".

Experimental design

1. The content of the paper is in line with the aims and scope of the journal.

2. The literature review has been performed using a systematic methodology: PRISMA framework ensures rigour. The method is described with a good level of detail.

3. Motivation for Excluding Sources:
The authors should justify the exclusion of conference proceedings and book chapters, addressing quality or scope concerns.

Validity of the findings

1. The conclusions are well-stated, directly addressing the original research questions. They summarise key findings and link them to the stated goals in the introduction. However:
- The conclusion does not clearly identify specific gaps or unresolved questions.
- While the authors mention future research, their suggestions remain broad. More concrete directions, such as testing specific theoretical integrations or exploring adaptive gamification for diverse learner groups, would be valuable.

2. Authors provide the details on the review methodology and processes ensuring ease of replication

Additional comments

To comply with PeerJ standards, do not acknowledge funders in the Acknowledgments section, as a separate Funding Statement is provided for that.

Cite this review as

---

## Round 0.2 · Minor Revisions

Experts have now judged your revised manuscript and as you can read, their comments require another revision. In particular, the reviewers ask (1) to better clarify parts of the methodology used during the literature review, (2) to revise or better justify the inclusion and exclusion criteria, (3) to add a "threats to validity" section, (4) to improve the language and future work, and (5) to provide a means to distinguish the studies/articles included in the review, not necessarily in the References. Please carefully revise your manuscript according to the comments from the reviewers or provide a rebuttal.

**Language Note:** The review process has identified that the English language must be improved. PeerJ can provide language editing services - please contact us at [email protected] for pricing (be sure to provide your manuscript number and title). Alternatively, you should make your own arrangements to improve the language quality and provide details in your response letter. – PeerJ Staff

Reviewer 1 ·

Basic reporting

The authors have replied to most of the comments received during the first review phase with attention, improving both the clarity and structure of the manuscript. To be ready for a publication I suggest to solve the following minor issues to further enhance the quality of the paper:

1) The methodology used for extracting and organizing themes during the review process should be clarified. Providing more details on how data was coded, how thematic consistency was ensured, and how disagreements among researchers were resolved would enhance the transparency and replicability of the study.
2) The manuscript would benefit from tighter language in several sections, particularly in the Discussion, where some paragraphs are overly long and occasionally redundant; refining these would improve the overall readability and flow.
3) While the authors rightly emphasize the strengths of the FSLSM framework, they should avoid presenting it as categorically superior to other models like HEXAD or MBTI without sufficient comparative empirical evidence, and instead acknowledge that different models may be suited to different contexts. Additionally, it would be helpful to introduce the theoretical framework earlier in the paper, perhaps at the end of the introduction or beginning of the discussion, to provide readers with a guiding structure.
4) A dedicated limitations section should be added to openly acknowledge the exclusion of certain types of literature (e.g., conference papers, book chapters), the lack of empirical validation of the proposed framework, and potential constraints in terms of generalizability.
5) Finally, the conclusion could be strengthened by specifying concrete future research directions, such as empirical testing of the proposed model in diverse educational contexts, longitudinal studies, or mixed-method evaluations involving direct learner input.

Experimental design

Comments already included in the basic report

Validity of the findings

Comments already included in the basic report

Additional comments

Comments already included in the basic report

Cite this review as

Reviewer 2 ·

Basic reporting

no comment

Experimental design

no comment

Validity of the findings

no comment

Additional comments

1) Concerning my earlier suggestion: “References (both the included studies and the additional references) are presented in a single list. However, having two separate lists would enhance transparency…”:
While the authors provide a rationale for not separating excluded studies and clarify that all sources in the Results and Discussion stem from included studies, this does not fully address the concern. The request was not about excluded studies, but about clearly distinguishing included studies from background references in the final reference list. Currently, readers cannot easily identify which sources were part of the systematic review. This leaves the burden on the reader to cross-reference manually. Listing included studies separately would significantly enhance the transparency and usability of the review, while not introducing redundancy.

2) Concerning my concerns about the inclusion criteria: While I appreciate the authors’ rationale for prioritising peer-reviewed journal articles, I would like to clarify that many high-ranked conferences undergo rigorous review processes that are comparable in quality to those of journals. Therefore, the exclusion of conference proceedings on the basis of peer review quality may not be fully justified. A more nuanced justification (such as evaluating conference proceedings based on the venue or the methodological rigour of individual contributions) would strengthen the authors’ argument, rather than assuming that all non-journal sources lack sufficient quality.

Cite this review as

---

## Round 0.3 · accepted · Accept

As you can read, both reviewers conclude that you have thoroughly revised your manuscript, addressing all of the reviewers' comments in a satisfactory manner. I agree with the reviewers that your manuscript is now ready for publication.

Reviewer 1 ·

Basic reporting

The article has been thoroughly revised, and the authors have provided detailed and thoughtful responses to all the comments raised by the reviewers. The revisions have significantly improved the clarity and quality of the manuscript. In my opinion, the paper now meets the necessary standards and is ready for publication.

Experimental design

The article has been thoroughly revised, and the authors have provided detailed and thoughtful responses to all the comments raised by the reviewers. The revisions have significantly improved the clarity and quality of the manuscript. In my opinion, the paper now meets the necessary standards and is ready for publication.

Validity of the findings

The article has been thoroughly revised, and the authors have provided detailed and thoughtful responses to all the comments raised by the reviewers. The revisions have significantly improved the clarity and quality of the manuscript. In my opinion, the paper now meets the necessary standards and is ready for publication.

Cite this review as

Reviewer 2 ·

Basic reporting

no comment

Experimental design

no comment

Validity of the findings

no comment

Additional comments

After reviewing the revised manuscript and the rebuttal letter, I find that all of my previous concerns have been clearly addressed and satisfactorily resolved. I therefore recommend accepting the paper in its current form.

Cite this review as